# Mast cell extracellular trap formation underlies vascular and neural injury and hyperalgesia in sickle cell disease

Donovan A Argueta[1],*, Huy Tran[2],*, Yugal Goel[1], Aithanh Nguyen[2], Julia Nguyen[2], Stacy B Kiven[1], Chunsheng Chen[2], Fuad Abdulla[2], Gregory M Vercellotti[2], John D Belcher[2], Kalpna Gupta[1,2]

**Sickle cell disease (SCD) is the most common inherited monogenetic disorder. Chronic and acute pain are hallmark features of SCD involving neural and vascular injury and inflammation. Mast cells reside in the vicinity of nerve fibers and vasculature, but how they influence these structures remains unknown. We therefore examined the mechanism of mast cell activation in a sickle microenvironment replete with cell-free heme and inflammation. Mast cells exposed to this environment showed an explosion of nuclear contents with the release of citrullinated histones, suggestive of mast cell extracellular trap (MCET) release. MCETs interacted directly with the vasculature and nerve fibers, a cause of vascular and neural injury in sickle cell mice. MCET formation was dependent upon peptidylarginine deiminase 4 (PAD4). Inhibition of PAD4 ameliorated vasoocclusion, chronic and acute hyperalgesia, and inflammation in sickle mice. PAD4 activation may also underlie neutrophil trap formation in SCD, thus providing a novel target to treat the sequelae of vascular and neural injury in SCD.**

## Introduction

Pain, inflammation, and oxidative stress are hallmark features of sickle cell disease (SCD), the most commonly inherited blood disorder (Kato et al, 2018). The cause of SCD was recognized ~3 quarters of a century ago as a point mutation in the beta-globin chain of RBCs leading to hemoglobin (Hb) polymerization and subsequent RBC sickling (Ingram, 1956). However, the complex pathobiology and sterile inflammation replete with *"cytokine storm"* pose a major challenge for treatment of SCD pain (Sundd et al, 2019). Opioids remain the common treatment for both acute vasoocclusive crises (VOC) and chronic pain in SCD (Carroll et al, 2016; Tran et al, 2017). However, morphine potentiates mast cell activation via MAS-related G protein–coupled receptor X2, which is

also a receptor for substance P (SP), a neuropeptide involved in immune cell recruitment, neurogenic inflammation, and pain when released from activated mast cells (Vincent et al, 2013; Gupta & Harvima, 2018; Navines-Ferrer et al, 2018; Green et al, 2019). SP levels are significantly higher in persons and mice with SCD, which may further amplify mast cell activation (Michaels et al, 1998; Kohli et al, 2010; Vincent et al, 2013; Brandow et al, 2016). However, the primary cause of mast cell activation in SCD remains unknown.

Mast cells reside in the vicinity of epithelial lining, blood vessels, and nerve fibers (Kunder et al, 2011; Gupta & Harvima, 2018; Elieh Ali Komi et al, 2020). Thus, mast cell activation may underlie vascular dysfunction, neural injury, and thinning of the epithelial layer in sickle mice (Hebbel & Vercellotti, 1997; Kohli et al, 2010; Vincent et al, 2013). Mast cells contribute to inflammation, neuroinflammation, and pathobiology of pain (Vincent et al, 2016; Mai et al, 2021; Bao et al, 2023). Upon activation, mast cells release tryptase and histamine and trigger nociceptors to release neuroinflammatory and vasoactive SP, which are elevated in the blood of persons with SCD (Michaels et al, 1998; Biedrzycki et al, 2009; Brandow et al, 2016; Allali et al, 2019; Pejler et al, 2022). Similar to neutrophils, mast cells also form extracellular traps (MCETs), an explosion of web-like decondensed chromatin with nuclear and mitochondrial DNA and citrullinated histones (Gupta & Harvima, 2018; von Kockritz-Blickwede et al, 2008). Peptidylarginine deiminase 4 (PAD4) catalyzes the citrullination of histone H3 by converting arginine to citrulline leading to the decondensation of nuclear chromatin (Fuhrmann et al, 2015).

The gene expression of *Padi*4 is elevated in subjects with SCD (Hounkpe et al, 2021), and neutrophil extracellular traps (NETs) contribute to vasoocclusion in murine and human SCD (Caudrillier et al, 2012; Chen et al, 2014; Vats et al, 2017). MCETs and NETs can cause tissue damage, which may be more pronounced in neighboring vascular and neural structures. However, the cause and contribution of MCETs to vascular and neural damage underlying VOC and chronic SCD pain are not understood. We therefore examined the novel process of MCET formation and regulation by PAD4, and their contribution to inflammation, vascular function,

[1]Division of Hematology/Oncology, Department of Medicine, University of California, Irvine, Irvine, CA, USA    [2]Division of Hematology, Oncology, and Transplantation, School of Medicine, University of Minnesota, Twin Cities, Minneapolis, MN, USA

Correspondence: kalpnag@hs.uci.edu
*Donovan A Argueta and Huy Tran contributed equally to this work

and acute and chronic pain in SCD using the well-established humanized HbSS-Berkeley (BERK) "sickle" mouse model that shows the characteristic features of sickle cell pathobiology and pain (hyperalgesia). HbSS-BERK mice are deleted for mouse $α$- and $β$-globins with the expression of >99% human sickle hemoglobin (Hb), whereas their genotype-matching control mice are also deleted for mouse $α$- and $β$-globins and exclusively express normal human HbA (Pászty et al, 1997; Sagi et al, 2018).

# Results and Discussion

### Sickle microenvironment stimulates MCET formation

Earlier findings from our laboratory had evinced the release of extracellular traps from mast cells and their interaction with nerve fibers and vasculature in the skin of HbSS-BERK mice (Gupta & Harvima, 2018). The citrullinated histones and explosive contents of the mast cell traps permeated the vasculature and were wrapped around the nerve fibers. We demonstrate that this novel phenomenon of MCET formation is incited by a sickle cell microenvironment replete with cell-free heme and inflammation. Using in vivo studies in HbSS and HbAA mice and in vitro studies using mast cells derived from the skin of HbSS and HbAA mice, we demonstrate that the PAD4-specific inhibitor, GSK484, attenuates MCET formation, acute and chronic hyperalgesia, and vaso-occlusion. Compelling evidence suggests that mast cells—tissue-resident granulocytes that reside near vasculature and nerve fibers—create a noxious microenvironment comprising inflammation, release of proteases, and neurogenic inflammation, leading to neural activation and pain in many preclinical and clinical studies including HbSS mice (Vincent et al, 2013; Gupta & Harvima, 2018; Green et al, 2019; Aguilera-Lizarraga et al, 2021; Meloto et al, 2021; Guida et al, 2022). In addition to the release of preformed products from the granules, mast cells release their contents through de novo synthesis of cytokines, nanotubes, extracellular vesicles, and traps that make direct contact with the neural and vascular structures (Gupta & Harvima, 2018; St John et al, 2023). Mast cell degranulation and vascular and peripheral nerve fiber injury have been observed in HbSS mice (Kohli et al, 2010; Vincent et al, 2013; Sadler et al, 2019). In addition to degranulation, mast cells release additional noxious substances including DNA and citrullinated histones forming MCETs, which can damage the nerve fibers and vasculature contributing to neuropathic and VOC pain. We therefore examined whether MCETs cause direct injury to the vasculature and nerve fibers in the skin and whether the sickle microenvironment promotes MCET formation.

We observed MCETs in the skin of HbSS mice demonstrated by the dense extracellular explosion of tryptase (red) and c-Kit (green) interspersing and scattered around nerve fibers (cyan) and clutching the blood vessels (magenta) in the skin of HbSS mice (Fig 1A and B), which was absent in control HbAA mice. We observed costaining of tryptase granules (red) in close proximity to nerve bundles (cyan) suggestive of mast cell activation. This is similar to earlier findings showing dense clusters of tryptase granules intercepting the nerve bundles in the skin of HbSS-BERK mice (Gupta & Harvima, 2018). Our observations are complemented by

other models of pain including mast cell degranulation evoked by glyceryl trinitrate–induced headache in rats and mast cell accumulation at the site of incision near the nerve fibers in the hind paw of mice in a post-surgical model of pain (Pedersen et al, 2015; Green et al, 2019). Similarly, tryptase (red), colocalized (yellow) with c-Kit (green), permeated the skin vasculature (CD31$^+$ cells, magenta) and might therefore boost vasculopathy in SCD. We next examined whether a pro-inflammatory environment and free heme released during hemolysis in SCD contribute to MCET formation. GSK484 treatment ameliorated MCET formation in the skin. Tryptase release and c-Kit expression are not observed/minimally expressed near the nerve fibers and blood vessels after GSK484 treatment of sickle mice (Fig 1A and B). We incubated mast cells isolated from the skin of HbAA and HbSS mice with (i) vehicle, (ii) 1 ng/ml TNF-$α$ for 4 h, (iii) 40 $μ$M hemin for 2 h, or (iv) 1 ng/ml TNF-$α$ for 2 h followed by 40 $μ$M hemin and TNF-$α$ for an additional 2 h (Fig 1C). HbSS and HbAA mouse–derived cutaneous mast cells incubated with both TNF-$α$ and hemin showed eruption of extracellular traps characterized by the release of SYTOX-labeled DNA extravasating from their nuclei with the appearance of spider web–like structures indicated by citrullinated histones (red) and extranuclear chromatin with nucleic acid structures (blue) (Fig 1D). Extracellular histones have been shown to have a cytotoxic effect and contribute to renal and liver injury via TLR4 and TLR2 (Xu et al, 2011; Allam et al, 2012), ischemia/reperfusion injury (Savchenko et al, 2014), platelet activation, and thrombosis (Fuchs et al, 2011), to promote anemia by inducing erythrocyte aggregation, Hb uptake in the spleen, and RBC sequestration and fragility (Kordbacheh et al, 2017). Notably, PAD4, which drives the citrullination and hence the extracellular presence of histones, has been suggested to play a significant role in neutrophil-driven thromboinflammation (Ansari et al, 2023).

MCETs extend tubular assemblies, which can reach longer distances and transport the cell's nuclear contents. Cutaneous mast cells from HbSS and HbAA mice showed a significant increase in the MCET length after treatment with TNF-$α$ or hemin alone ($P < 0.001$), but the combination had a greater effect than either TNF-$α$ or hemin alone (Fig 1E). Interestingly, mast cells from HbSS mice incubated with TNF-$α$/hemin showed significantly greater MCET length than HbAA mast cells ($P = 0.0035$), suggesting that mast cells from HbSS mice may be primed for a greater response. We speculate that this interception of nerve fibers with mast cell tryptase may lead to neural injury, which may contribute to neuropathic pain. It is likely that mast cell activation contributes to the peripheral neural injury with sprouting nerve fibers in the skin, myelin sheath instability in the sciatic nerve, and neurovascular permeability that occur constitutively in HbSS mice (Kohli et al, 2010; Vincent et al, 2013; Sadler et al, 2019).

HbSS mast cells express significantly higher *Tlr4* and *Fcer1a* transcripts compared with HbAA mast cells (Vincent et al, 2013), which may lead to downstream activation of PAD4, followed by citrullination of histones, a critical step in extracellular release of nuclear contents (Fuhrmann et al, 2015). TLR4 has been widely demonstrated to play a role in NET formation (Clark et al, 2007; Tsourouktsoglou et al, 2020), whereas Fc$ε$R1 negatively regulates microtubule assembly contributing to the degranulation of mast cells (Kraft & Kinet, 2007); however, their role in MCET formation remains unclear. We therefore evaluated the effect of TLR4

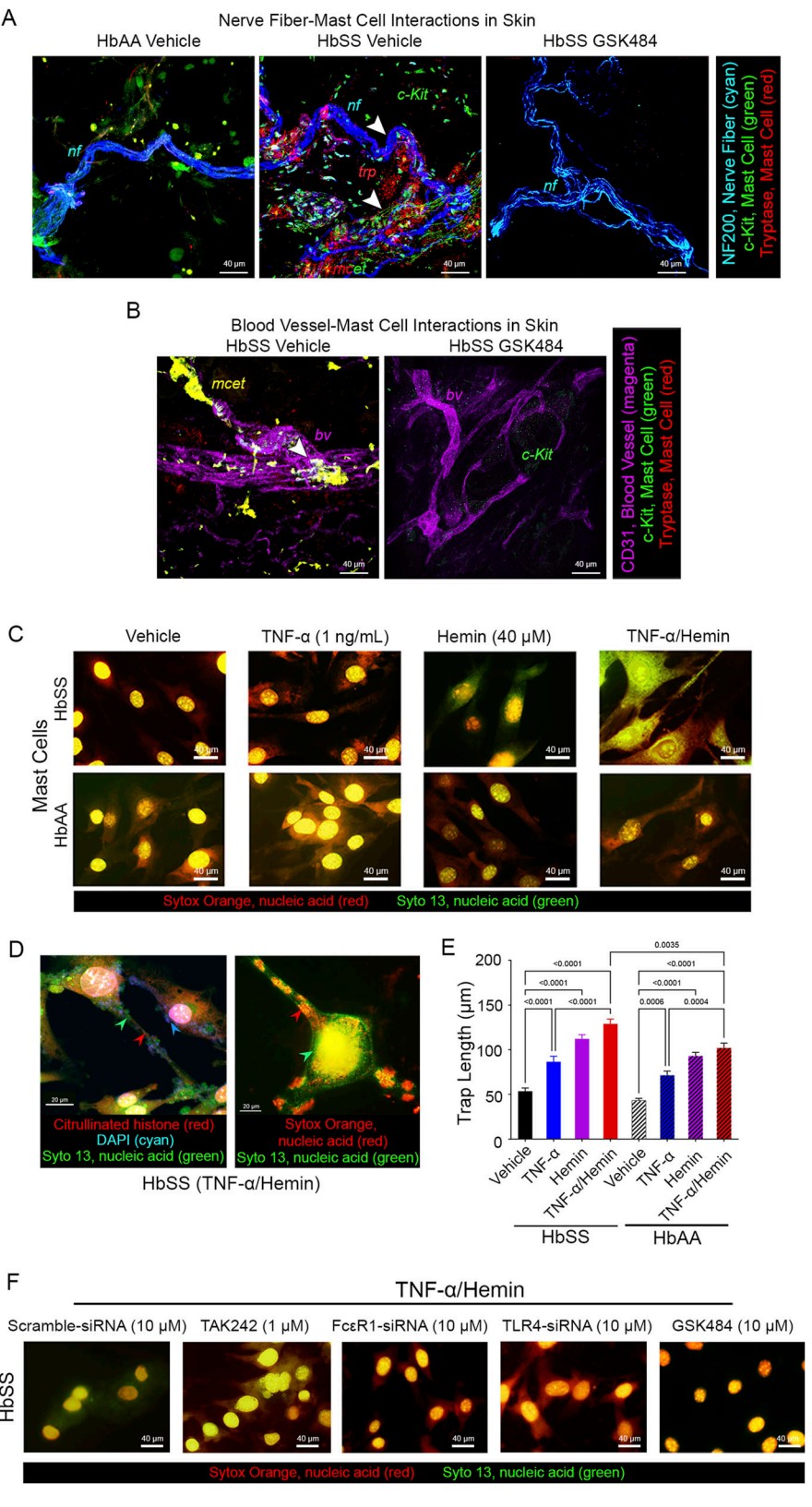

**Figure 1. Sickle microenvironment drives mast cell extracellular trap (MCET) release.**
**(A)** Dorsal skin of female control HbAA mice treated with the vehicle and sickle HbSS mice treated with the vehicle or GSK484 (sc, 20 mg/kg/day) was imaged with a laser scanning confocal microscope in 100-μm immunostained (see the Materials and Methods section for details) sections. *Z*-stacks of 1.0 μm were sequentially acquired at 60x magnification. Association of mast cells with vasculature and nerve bundles is shown: nerve bundles (cyan, anti-neurofilament H-200) surrounded by degranulating mast cells (green, anti-c-Kit) loaded with tryptase granules (red, anti-tryptase). Mast cells are extending green nanotubes interspersing and disrupting the nerve bundles, and tryptase granules are intertwined around the nerve bundles, suggestive of MCET formation in HbSS mice. **(B)** Mast cells, indicated with c-Kit and tryptase, associate with CD31⁺ (magenta) endothelial cells in HbSS skin. White arrows indicate mast cell interactions with nf or bv. **(C)** Incitement of a sickle microenvironment with TNF-α (1 ng/ml), hemin (40 μM), or TNF-α/hemin caused MCET formation in mast cells cultured from skin of HbSS female mice, shown with the formation of rigid DNA fibers (stained with cell-permeable DAPI and SYTO 13 and cell-impermeable SYTOX Orange). **(D)** HbSS mast cells treated with TNF-α/hemin showed citrullinated histones (red) and DNA-binding fluorescent probes, DAPI (cyan), SYTO 13 (green), and SYTOX Orange (red), indicating an extranuclear DNA that overlaps with histone citrullination. **(E)** MCETs were elongated in HbSS and HbAA mast cells with TNF-α, hemin, or combined treatment, and to a greater extent in HbSS mast cells compared with TNF-α or hemin alone. The data are analyzed with regular two-way ANOVA with Tukey's post hoc multiple comparisons test. Data shown are the mean ± SEM of N = 17–24 cells. **(F)** TNF-α/hemin-induced MCET formation was abrogated by pretreatment with scrambled siRNA (10 μM), the TLR4 inhibitor TAK242 (1 μM), siRNA silencing of FcεR1 and TLR4 (10 μM each), and PAD4 inhibition with GSK484 (10 μM). Each image represents multiple images from five different 5.0-mo-old female mice. Abbreviations: bv, blood vessel; MCET, mast cell extracellular trap; nf, nerve fiber; PAD4, peptidylarginine deiminase 4; TLR4, Toll-like receptor 4; TNF-α, tumor necrosis factor alpha; trp, tryptase.

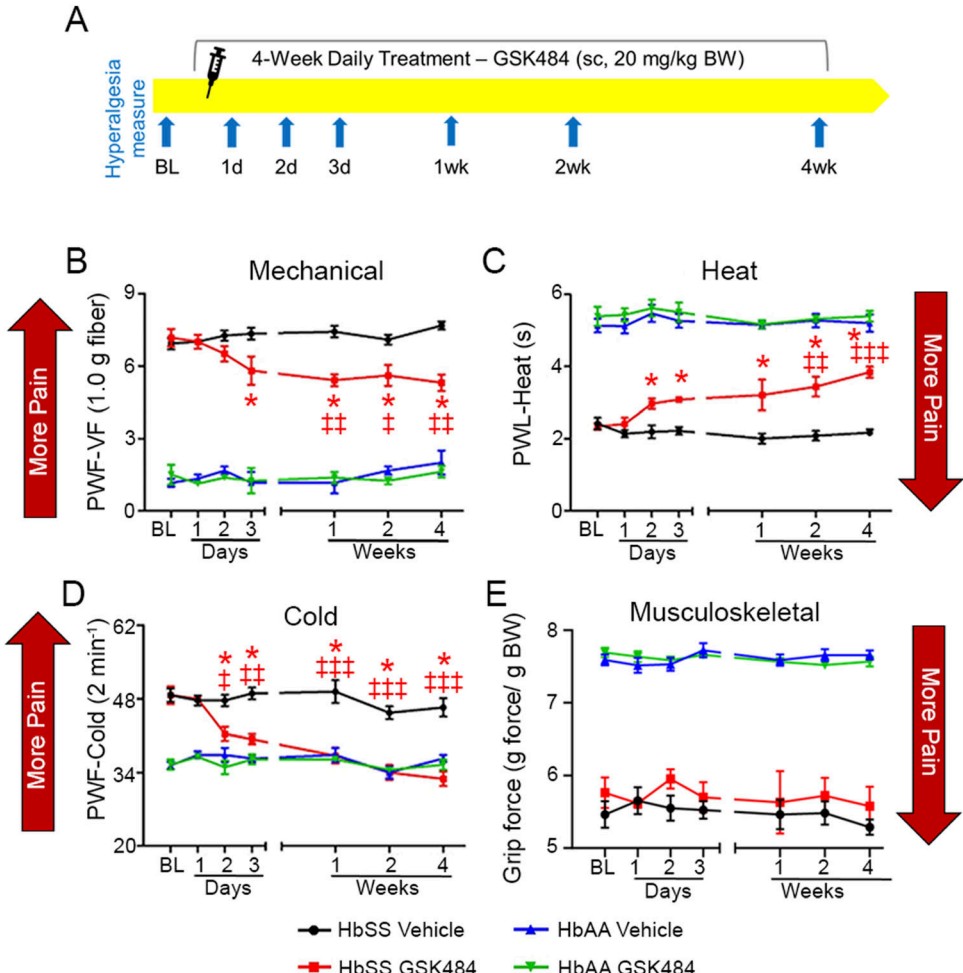

**Figure 2. Peptidylarginine deiminase 4 (PAD4) inhibition reduces chronic hyperalgesia in HbSS mice.**
**(A)** HbSS and HbAA mice were treated daily with the PAD4 inhibitor, GSK484 (sc, 20 mg/kg body weight/day), over a 4-wk period. Hyperalgesia was analyzed at baseline (BL) before starting the treatments, and after the treatments on days 1–3 and at weeks 1, 2, and 4. **(B, C)** PAD4 inhibition significantly reduced mechanical hyperalgesia compared with vehicle treatment after 3 d of treatment until the end of the treatment regimen, showing a significant reduction from baseline from 1 wk onward, similar to (C) heat hyperalgesia, which was improved from day 2 onward compared with the vehicle and week 2 onward compared with baseline. **(D)** Cold hyperalgesia showed the greatest response, indicating less hyperalgesia compared with the vehicle and baseline from day 2 onward. **(E)** Spontaneous, non-evoked musculoskeletal hyperalgesia was not affected by PAD4 inhibition. Data are shown as the mean ± SEM and analyzed with regular two-way ANOVA with Tukey's post hoc multiple comparisons test. * indicates a significant difference between HbSS vehicle and HbSS GSK484 groups at matching time points. ‡ indicates a significant difference in the treatment group compared with BL. *,‡$P < 0.05$, ‡‡$P < 0.01$, ‡‡‡$P < 0.001$. N = 5–6. ~5-mo-old female HbSS mice. Abbreviations: BL, baseline; BW, body weight; S.C., subcutaneous; PWF, paw withdrawal frequency; PWL, paw withdrawal latency; VF, von Frey.

inhibition with TAK242, siRNA silencing of TLR4 and FcεR1, and PAD4 inhibition with GSK484 on TNF-α/hemin-induced MCET formation in HbSS mast cells (Fig 1F). TAK242 (1 μM) and siRNA silencing of TLR4 or FcεR1 prevented TNF-α/hemin-induced MCET formation indicated by decreased SYTO 13 and SYTOX Orange dye compared with TNF-α/hemin-stimulated mast cells (Fig 1F). It is likely that TLR4 signaling, inflammation, and ROS released by heme stimulation mediate the activation of mast cells and downstream PAD4 activation leading to MCET formation. Heme-induced TLR4 signaling contributes to vascular stasis (Belcher et al, 2014; Beckman et al, 2020). PAD4 inhibition with GSK484 (10 μM) completely blocked TNF-α/hemin-induced MCET formation in HbSS mast cells (Fig 1F). Compared with TLR4 and FcεR1 inhibition, PAD4 inhibition more prominently reduces DNA extravasation, suggesting a necessary and sufficient role of PAD4 in MCET formation in a sickle microenvironment. Collectively, these in vitro observations suggest a crosstalk between multiple receptors and their pathways, which converge at the downstream activation of PAD4, and PAD4 catalysis of histone citrullination is necessary for MCET formation, which is another feature with similarity to NET formation (Tsourouktsoglou et al, 2020). We found that the explosive nuclear and intracellular contents that are spread through expansion of nanotubes and

traps are able to directly interact with and permeate the vascular and neural structures, which may underlie neuropathic pain and vascular dysfunction observed in SCD. Neuropathic pain in SCD is extremely challenging to treat and may continue even after transformative bone marrow transplant (Darbari et al, 2019; Krishnamurti et al, 2024). Thus, prevention and treatment of MCET formation may have an ameliorating effect on chronic neuropathic pain in SCD.

### PAD4 inhibition reduces chronic hyperalgesia in HbSS mice

HbSS mice display features of chronic hyperalgesia compared with HbAA mice (Kohli et al, 2010; Hillery et al, 2011), but to date, extracellular trap formation has not been evaluated in the context of SCD pain. Daily treatment with the PAD4 inhibitor, GSK484, subcutaneously (sc, 20 mg/kg/d) significantly reduced thermal and mechanical hyperalgesia starting on days 2 and 3, respectively, which were sustained or further reduced throughout the 4-wk treatment period in HbSS mice compared with pretreatment baseline (BL, $P < 0.05$) (Fig 2A–D). HbAA mice treated with GSK484 (sc, 20 mg/kg/d) showed no change over a 4-wk treatment period. Inhibition of cold hyperalgesia with GSK484 is most remarkable in

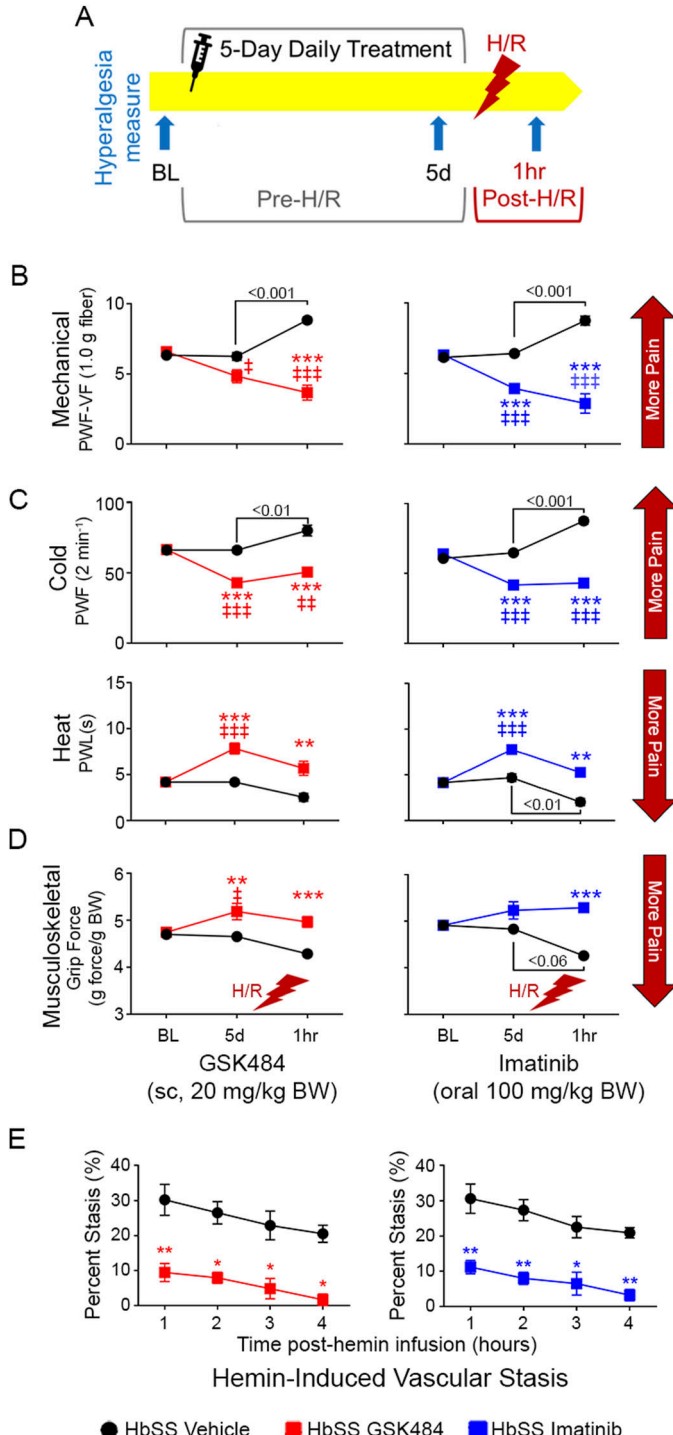

HbSS mice compared with mechanical hyperalgesia as it reaches the same levels as HbAA mice that do not display cold hyperalgesia. No changes were observed with musculoskeletal hyperalgesia (Fig 2E), which is likely arising from avascular necrosis that requires therapies to restore the damaged bone tissue (De Franceschi et al, 2020). The anti-hyperalgesic effect of PAD4 inhibition was sustained over the 4-wk treatment period in HbSS mice, without causing tolerance. These findings suggest constitutive PAD4 activity, leading to MCET formation in vivo, may contribute to the persistence of chronic mechanical and thermal hyperalgesia in SCD. Persistence of mechanical and cold allodynia and hyperalgesia is a critical feature of pain in SCD, which may drive central sensitization (Brandow et al, 2016; Ezenwa et al, 2018; Molokie et al, 2020). A multicenter study of ~300 SCD patients over a period of 3 yr reported that "colder temperatures exacerbated sickle cell pain" (Smith et al, 2009). Cold hypersensitivity is a life-threatening condition in children and suggested to be a cause of acute pain in SCD (Ivy et al, 2023). Mechanical hyperalgesia is another characteristic feature of SCD with increased sensitivity to touch and even wind speed (Nolan et al, 2008). These findings suggest that PAD4 inhibition directly acts upon mast cells interacting with sensory nerve fibers involving transient receptor potential vanilloid (TRPV) channels, which regulate thermal sensitivity. Mast cell activation leads to thermoregulatory neural circuit activation via protease-activated receptors and TRPV1-causing hypothermia (Bao et al, 2023). It is likely that GSK484 also inhibits NETs (Lewis et al, 2015; Wang et al, 2023), which are implicated in VOC (Barbu et al, 2020). Thus, PAD4 targeting offers a novel approach to target pain-relieving and/or pain-preventing strategies.

### Inhibition of PAD4 and mast cell activation reduces acute VOC-like pain and vascular stasis in HbSS mice

NETs have been suggested to be associated with VOC in mice and persons with SCD (Chen et al, 2014). However, their impact on acute pain of VOC has not been examined. MCET inhibition may be a target for preventing VOC-related pain by preventing vascular dysfunction (i.e., stasis). Incitement of hypoxia/reoxygenation (H/R, 3 h at 8% O₂, followed by 1 h at room air/normoxia) simulates VOC-like pain (hyperalgesia) in HbSS mice. Reperfusion injury is a potent driver of perivascular mast cell activation that may underlie pain and vascular dysfunction (Fig 3A) (Reichel et al, 2011; Cain et al, 2012). In three separate studies, imatinib, an inhibitor of mast cells via *c-Kit*, showed amelioration of VOC in subjects with SCD without impacting the hematologic profile or fetal Hb (Close & Lottenberg, 2009;

**Figure 3. Inhibition of peptidylarginine deiminase 4 and mast cell activation attenuate acute hyperalgesia and microvascular stasis.**
**(A, B)** Schema showing daily treatment with GSK484 (sc, 20 mg/kg/d) or imatinib (oral, 100 mg/kg/d) over a 5-d period, with assessment of hyperalgesia at baseline, day 5 before incitement of hypoxia/reoxygenation (H/R, 3 h at 8% O₂, followed by 1 h at ~20% O₂, normoxia), and 1 h after H/R. GSK484 and imatinib treatments showed a significant reduction in hyperalgesia: **(B)** mechanical hyperalgesia, indicated by a reduction in paw withdrawal frequency (PWF) compared with baseline before H/R (*P* < 0.05), and significantly lower PWF after H/R when compared to vehicle treatment (*P* < 0.01). **(C)** Cold hyperalgesia was similarly improved with treatments, indicated by reduced PWF and increased

paw withdrawal latency. **(D)** Spontaneous, non-evoked musculoskeletal hyperalgesia was significantly improved by treatments after H/R compared with the vehicle (*P* < 0.001). **(E)** Hemin infusion induces stasis in HbSS mice, which was significantly reduced (*P* < 0.05) in mice pretreated for 5 d with GSK484 and imatinib compared with the vehicle. Data are shown as the mean ± SEM and analyzed with regular two-way ANOVA with Tukey's post hoc multiple comparisons test. * indicates a significant difference between HbSS vehicle and HbSS treatment groups at matching time points. ‡ indicates a significant difference in the treatment group compared with BL. *,‡*P* < 0.05, **,‡‡*P* < 0.01, ***,‡‡‡*P* < 0.001. N = 6–8. ~5-mo-old female HbSS mice. Abbreviations: BL, baseline; BW, body weight; PWF, paw withdrawal frequency; PWL, paw withdrawal latency S.C., subcutaneous; VF, von Frey.

Stankovic Stojanovic et al, 2011; Karimi et al, 2023). Thus, we evaluated the effect of PAD4 and mast cell inhibition with GSK484 and imatinib, respectively, on H/R-induced acute hyperalgesia and vascular stasis. Mechanical and thermal hyperalgesia were significantly increased in vehicle-treated HbSS mice, 1 h after incitement of H/R ($P < 0.001$ and $P < 0.001$, respectively, Fig 3B and C). Pretreatment for 5 d with GSK484 (sc, 20 mg/kg/d) or imatinib (oral gavage, 100 mg/kg/d) significantly reduced mechanical and thermal hyperalgesia compared with baseline (BL; $P < 0.05$ and $P < 0.001$, respectively, Fig 3B and C) and prevented H/R-induced acute mechanical, thermal, and musculoskeletal hyperalgesia, which were significantly improved compared with the vehicle ($P < 0.01$, Fig 3B–D). Vascular occlusion underlies stasis, a key feature of SCD contributing to ischemia/reperfusion injury and VOC pain (Kalambur et al, 2004; Belcher et al, 2014). Stasis was measured in HbSS mice implanted with dorsal skin-fold chambers (DSFCs) after incitement of stasis with intravenous hemin infusion (iv, 1.6 $\mu$mol/kg). Pretreatment for 5 d with GSK484 or imatinib significantly reduced stasis over a 4-h incitement period compared with vehicle-treated HbSS mice ($P < 0.05$ and $P < 0.01$, respectively, Fig 3E), indicating greater blood flow that suggests amelioration of VOC-like processes (Federti et al, 2023). Thus, inhibition of MCET formation with GSK484 and attenuation of mast cell activation with imatinib ameliorate vascular stasis, as well as acute hyperalgesia, suggesting a contribution of mast cell activation and MCET formation in acute VOC pain in SCD. Inhibition of PAD4 ameliorates constitutive hyperalgesia, as well as vascular stasis, and H/R-evoked acute hyperalgesia in HbSS mice, demonstrating the contribution of MCETs in chronic and acute pain in SCD. Notably, we show that TNF-$\alpha$ and hemin, which are abundant in tissues and circulation in SCD, synergize to promote MCET formation. The mechanism of MCET formation has similarities to NETs, which involves heme in a pro-inflammatory, cytokine-rich environment associated with the pathogenesis of VOC in SCD (Chen et al, 2014). Thus, MCETs may contribute to both neuropathic pain, as well as vascular stasis, and acute VOC pain.

### PAD4 inhibition attenuates inflammation in HbSS mice

Sickle mast cell activation also drives inflammation and vaso-occlusion via other mechanisms, including neutrophil recruitment and increased P-selectin adhesion molecule expression (Kubes & Gaboury, 1996; Torres et al, 2002; Turhan et al, 2002; Zhang et al, 2011; Pertiwi et al, 2019; Tran et al, 2019). Previous studies have implicated mast cell degranulation in P-selectin–dependent leukocyte rolling, recruitment, and adhesion (Ley, 1994; Gaboury et al, 1995; Drube et al, 2016). We previously found that conditioned medium from mast cells of HbSS mice enhanced endothelial P-selectin expression (Tran et al, 2019). The P-selectin–mediated adhesion of neutrophils and sickle RBCs underlies vasoocclusion and stasis leading to excruciating acute VOC pain (Manwani & Frenette, 2013). Furthermore, ROS formation and extravasation of neutrophils from perivascular mast cell activation may cause injury to the nerve fibers observed in HbSS mice (Kohli et al, 2010; Tran et al, 2019; Dudeck et al, 2021). After 4-wk GSK484 treatment (20 mg/kg/d; sc), HbSS mice showed a significant reduction in the circulating white blood cells (WBCs; $P = 0.0106$), serum concentration of pro-

inflammatory granulocyte–macrophage colony-stimulating factor (GM-CSF, $P < 0.001$), and number of degranulating cutaneous mast cells ($P < 0.001$) compared with vehicle treatment (Fig 4A–D). GM-CSF is associated with WBC production and decreases circulating fetal Hb concentrations and cAMP activity in SCD (Ikuta et al, 2011). Skin from GSK484-treated HbSS mice showed significantly reduced release of the monocyte chemoattractant protein 1 (MCP-1), TNF-$\alpha$, and regulated on activation, normal T-cell expressed and secreted protein (RANTES, $P = 0.0229$, $P = 0.0154$, and $P = 0.0290$, respectively), which are constitutively elevated in HbSS mice, compared with vehicle treatment (Fig 4E). MCP-1 may contribute to chronic SCD pain based on seminal observations in HbSS mice, and RANTES, also released from HbSS mast cells, is involved in the recruitment of leukocytes (Appay & Rowland-Jones, 2001; Vincent et al, 2013; Sadler et al, 2018). TNF-$\alpha$ is a "*sentinel cytokine*," but its pleiotropic effects can contribute to multiple inflammatory pathologies. The significance of inhibiting TNF-$\alpha$ release from mast cells is many-fold. In addition to costimulating mast cell activation, TNF-$\alpha$ is also implicated in the neutrophil influx (Malaviya et al, 1996). Adverse effects of TNF-$\alpha$ in sickle cell vasoocclusion are well known (Solovey et al, 2017). Earlier studies from our group found that the blockade of TNF-$\alpha$ activity with etanercept attenuated the elaboration of cytokines MCP-1 and IL-6, vascular cell adhesion molecule 1 (VCAM-1), and related neuropeptides, but it did not reduce hyperalgesia (Solovey et al, 2017). Amelioration of both chronic pain, as well as stasis, and H/R-incited acute pain with GSK484 may involve these downstream effector cytokines and cellular adhesion. These data support a novel role of PAD4-dependent MCET formation in the initiation and sustained activation of mast cells. A consequence of this aberrant signaling is apparent in SCD through vascular stasis, which arises through enhancement of adhesion molecules in endothelium and a pro-inflammatory milieu.

Herein, we present novel findings that MCET formation mediated by PAD4 contributes to chronic and acute pain, vascular stasis, and inflammation in SCD. NETs have been demonstrated in both mice and persons with SCD (Chen et al, 2014), but the cause of NET formation in SCD and its role in pain remained unknown. It is likely that PAD4 drives the NET formation in SCD similar to the activation of MCET formation. Notably, plasma PAD4 activity and expression are higher in SCD subjects with acute VOC versus healthy individuals without SCD (Hounkpe et al, 2021). Targeting PAD4-mediated extracellular trap formation may ameliorate both neutrophil and mast cell–mediated inflammation, vasoocclusion, and pain and may even have a disease-modifying effect. Therefore, PAD4 is a novel treatable target for developing therapeutics for ameliorating sickle cell pain.

# Materials and Methods

### Mouse model of SCD

We used humanized homozygous HbSS-BERK transgenic "sickle" mice with murine $\alpha$- and $\beta$-globin knockouts ($\alpha^{-/-}$, $\beta^{-/-}$), which express >99% human sickle Hb and demonstrate severe features of SCD and chronic mechanical, thermal, and musculoskeletal hyperalgesia (Pászty et al, 1997; Sagi et al, 2018). "Control" mice with

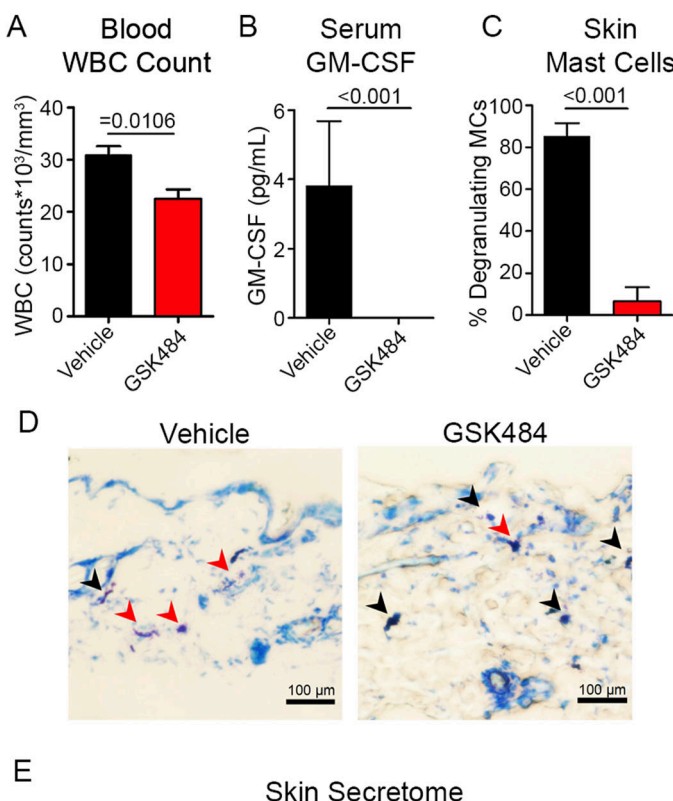

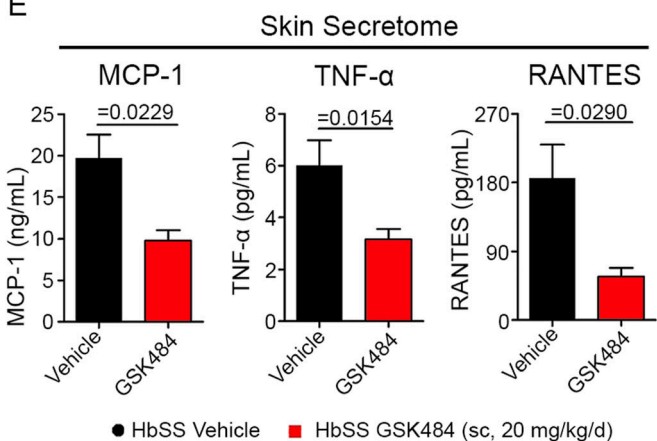

**Figure 4. Peptidylarginine deiminase 4 inhibition has anti-inflammatory benefits in HbSS mice.**

HbSS mice were treated daily for 4 wk with the vehicle or GSK484 (sc, 20 mg/kg/d), and blood, skin, and organs were collected after euthanasia. **(A, B, C)** GSK484 treatment significantly reduced the (A) circulating white blood cell counts ($P = 0.0106$), and (B) serum GM-CSF and (C) number of degranulating mast cells in dorsal skin, compared with vehicle treatment ($P < 0.001$). **(D)** Representative images of toluidine blue–stained skin biopsies; scale bar = 100 $\mu$m; yellow arrow: intact mast cell; red arrow: degranulating mast cell. **(E)** Secretome from dorsal skin biopsies revealed a significant reduction in (E) MCP-1, TNF-$\alpha$, and RANTES ($P = 0.0229$, 0.0154, and 0.0290, respectively) after GSK484 treatment compared with the vehicle. Data are shown as the mean $\pm$ SEM and analyzed with an unpaired two-tailed $t$ test. Female mice at ~5 mo of age; N = 6–8. Abbreviations: GM-CSF, granulocyte–macrophage colony-stimulating factor; MC, mast cell; MCP-1, monocyte chemoattractant protein 1; RANTES, regulated on activation, normal T-cell expressed and secreted protein; TNF-$\alpha$, tumor necrosis factor alpha; WBC, white blood cell.

the same mixed genetic background express normal human HbA (HbAA-BERK), without mouse $\alpha$- and $\beta$-globins (Pászty et al, 1997). Mice were bred and phenotyped in our laboratory as described

previously (Kohli et al, 2010; Sagi et al, 2018). Experiments were approved by Institutional Animal Care and Use Committees (IACUC) and complied with National Institutes of Health guidelines.

**Mast cell analysis in the skin**

Dorsal skin fixed in Zamboni's reagent was stained with chicken anti-*neurofilament H-200* (NF200, 1:1,000; RRID:AB_2149618, #ab72996; Abcam), rabbit anti-c-*Kit* (1:200; RRID:AB_2280836, #sc-5535; Santa Cruz Biotech), and goat anti-*tryptase* (1:200; RRID: AB_2256311, #sc-32473; Santa Cruz) (Gupta & Harvima, 2018). Colocalization of mast cells and vasculature was observed using antibodies for c-Kit and tryptase, described above, and endothelial cells with rat anti-*CD31/PECAM* (1:200; RRID:AB_627028, sc-18916; Santa Cruz). Sections were subsequently labeled with 2° anti-bodies at 1:400 dilution: Cy2-donkey anti-*chicken* (RRID:AB_2340370, #703-225-155; Jackson ImmunoResearch), Cy2-donkey anti-*rat* (RRID: AB_2340673, #712-225-150; Jackson), Cy3-donkey anti-*rabbit* (RRID: AB_2307443, #711-165-152; Jackson), and Cy5-donkey anti-*goat* (RRID:AB_2340415, #705-175-147; Jackson), correspondingly. *Z*-stacks of 1.0 $\mu$m step size were sequentially acquired using an Olympus FluoView FV1000 BX2 (Olympus) upright laser scanning confocal microscope at 60 X (numerical aperture 1.35) magnification, and images were pseudo-colored with ImageJ software (National Institutes of Health, Bethesda, MD).

**Preparation of reagents and inhibitors**

GSK484 was prepared with 10% DMSO in sterile saline and ad-ministered subcutaneously (sc, 20 mg/kg/d for 5 d). Imatinib mesylate (Novartis) was prepared in sterile PBS and administered by oral gavage (100 mg/kg/d for 5 d) (Huy Tran et al, 2019). Hemin (1.6 $\mu$mol/kg) was prepared by combining 10 mg hemin chloride (#H651-9; Frontier Scientific), 10 mg D-sorbitol (#S6021; Sigma-Aldrich), and 6.9 mg sodium carbonate (#S2127; Sigma-Aldrich) in 5.7 ml sterile saline for 30 min away from light, and then, it was diluted to a working concentration, passed through a 0.22-$\mu$m filter, and administered as a bolus via intravenous tail vein infusion for assessment of vascular stasis (Belcher et al, 2014).

**Isolation and culture of mast cells from skin**

Dorsal mouse skin was shaved before collection (Vincent et al, 2013). Mice were humanely euthanized with compressed $CO_2$ gas, and then, 1-cm² punch biopsies (~1 g total) were digested in sterile Dulbecco's PBS containing 0.2 mg/ml collagenase (#C9697; Sigma-Aldrich), 0.1 mg/ml hyaluronidase (#H3506; Sigma-Aldrich), and 0.2 mg/ml protease (#P8811; Sigma-Aldrich) at 37°C for 1 h followed by repeat wash/incubation for an additional 30 min, and filtered through a 70-$\mu$m cell strainer. The filtrate was placed into isotonic Percoll (#P1644; Sigma-Aldrich) solution and centrifuged for 20 min at 500$g$. Mast cells that passed through Percoll were collected, washed, and suspended in complete growth media (RPMI 1640 media containing 10% FBS, 100 U/ml penicillin/100 mg/ml strep-tomycin, 0.25 M Hepes, 1.2 mg/ml sodium bicarbonate, 4 mM L-glutamine, and 10 ng/ml recombinant mouse stem cell factor [#GF141; Sigma-Aldrich]). Purity was determined with toluidine blue

stain (#198161; Sigma-Aldrich) as previously described (Vincent et al, 2013; Vang et al, 2015; Campillo-Navarro et al, 2017; Tran et al, 2021). Mast cells were enriched by positive selection using anti-*CD117*-MicroBeads (RRID:AB_2753213, #130-091-224; Miltenyi Biotec) and magnetic separation on a MACS column (#130-095-691; Miltenyi Biotec).

## Mast cell activation and trap formation

Primary mast cells obtained from the skin of HbAA and HbSS mice were incubated in media containing 1% FBS for 2 h with 1 ng/ml TNF-*α* followed by the addition of 40 *μM* hemin for an additional 2 h (Huy Tran et al, 2019). Mast cell inhibitor/tyrosine kinase inhibitor imatinib mesylate (30 *μM*; Novartis), TLR4 inhibitor TAK242 (0.5 and 1.0 *μM*; #CLI-095; InvivoGen), and GSK484 (10 *μM*, #SML1658; Sigma-Aldrich) were all prepared fresh in complete media for cellular studies (Cerrato et al, 2010; Vincent et al, 2013, 2016; Tran et al, 2019; Lei et al, 2021). Mast cells were also transfected with FcεR1 siRNA (10 *μM*, #sc-45268; Santa Cruz), TLR4 siRNA (10 *μM*, #sc-40261; Santa Cruz), or scramble siRNA negative control (10 *μM*, # 4390843; Thermo Fisher Scientific) using Lipofectamine RNAiMAX Transfection Reagent (#13778100; Thermo Fisher Scientific) per the manufacturer's recommendation. Cells were washed with PBS and stained with cell-impermeable SYTOX Orange (1:1,000; #S11368; Thermo Fisher Scientific), cell-permeable SYTO 13 (1:1,000; #S7575; Thermo Fisher Scientific), and rabbit anti-*citrullinated histone H3* (1:1,000; RRID: AB_304752, #ab5103; Abcam), followed by Cy5-donkey anti-*rabbit* (1: 400; RRID:AB_2340607, #711-175-152; Jackson) secondary antibodies, and diamidino-2-phenylindole (DAPI, 1:2000; #62248; Thermo Fisher Scientific) nuclear stain. *Z*-stacks of 1.0 *μm* step size were acquired sequentially at 60 X (numerical aperture 1.35) magnification with a laser scanning confocal microscope.

## Pain-related behaviors

Mice were acclimatized to each test protocol in a quiet room at a constant temperature for 30 min and tested for mechanical, deep tissue/musculoskeletal (grip force), and thermal (cold) hyperalgesia. Mice were familiarized with the testing procedure by performing repeat measurements before baseline. Three measurements were recorded for each mouse per test unless specified, and there was an interval of 15 min between tests (Kohli et al, 2010; Sagi et al, 2018).

### Mechanical hyperalgesia
Mice were placed into glass enclosures (10 × 6.5 × 6.5 cm) on an elevated wire mesh. von Frey (Semmes–Weinstein) monofilaments (Stoelting Co) with 9.8 mN (1.0 g) calibrated bending forces were applied for 1–2 s or until the mouse withdrew its paw in response to the stimulus as previously determined. The paw withdrawal frequency (PWF) out of 10 applications was measured per hind paw, with higher PWF indicating more mechanical hyperalgesia.

### Grip force
To evaluate deep tissue/musculoskeletal hypersensitivity, the tensile force of peak forelimb exertion was measured using a computerized grip force meter (SA Maier Co.). During testing, each

mouse was held by its tail, gently passed over a wire mesh grid, and allowed to grip the wires with only their forepaws. The peak force exerted against the transducer was recorded in grams for three repetitions and averaged for each mouse. Grip force measurements were normalized by body weight (BW) in grams. A decrease in grip force is indicative of more musculoskeletal hyperalgesia.

### Heat hyperalgesia
A radiant heat stimulus was applied to the plantar surface of the hind paw from below with a projector lamp bulb (CXL/CXR, 8 V, 50 W) using PAW Thermal Stimulator (University Anesthesia Research & Development Group, University of California, San Diego, CA). Paw withdrawal latency to the nearest 0.1 s was recorded when the mouse withdrew its paw from the stimulus.

### Cold hyperalgesia
Mice were gently placed on a cold plate maintained at 4°C (#35100; Ugo Basile), and PWF during a 2-min period was recorded. Higher PWF indicates greater cold hyperalgesia. Cold hyperalgesia measurements are performed once per mouse because of extreme sensitivity to cold and risk of injury and death in HbSS mice (Brandow et al, 2013; Zappia et al, 2014).

## Assessment of vascular stasis

We used DSFCs to examine vasoocclusion in mice anesthetized with ketamine (106 mg/kg) and xylazine (7.2 mg/kg) (Kalambur et al, 2004; Vercellotti et al, 2016). HbSS mice were pretreated with the vehicle, imatinib (oral gavage, 5 mg/kg/d), or PAD4 inhibitor GSK484 (sc, 20 mg/kg/d) for 5 d. DSFCs were surgically implanted, and 20–25 flowing venules in the DSFC window of each mouse were selected and mapped using intravital microscopy using a Nikon microscope (model E400; Nikon Inc.) equipped with a Dage-MTI CCD television camera (model CCD-300T-RC; Dage-MTI Inc.) and a Sony U-matic video recorder (model VO5800; Sony). Immediately after baseline selection of flowing venules, mice were infused with hemin (1.6 *μmol/kg*). The same venules were then re-examined for vasoocclusion (no blood flow) at 1, 2, 3, and 4 h post-hemin infusion (Kalambur et al, 2004; Vercellotti et al, 2016).

## White blood cell analysis

At the endpoint, whole blood was collected via tail vein, combined with 100 mM ethylenediaminetetraacetic acid (EDTA), pH 7.5, in sterile PBS at a 2:1::blood:EDTA ratio, and assayed immediately. WBC counts were measured using Animal Blood Counter (abc Plus, Scilvet) (Vincent et al, 2013).

## Assessment of peripheral inflammation

At the endpoint of the study, mice were humanely euthanized and blood serum was collected for cytokine measurement with ELISA; dorsal skin punch biopsies (4 mm) were collected, fixed in Zamboni's reagent for mast cell assessment, or collected and incubated in culture media (RPMI 1640 with 100 U/ml penicillin + 100 mg/ml streptomycin) at 37°C with 5% $CO_2$ for 24 h for cytokine measurement with ELISA. Mast cells in skin sections were stained with

toluidine blue. In brief, the toluidine blue stain was prepared by dissolving 0.25 g toluidine blue (cat#01804; Chem-Impex International) in 35 ml distilled water, 15 ml ethanol 100%, and 1 ml HCl. After deparaffinization, the skin sections were incubated in toluidine blue for 1 min at room temperature, washed with distilled water, and air-dried. The stained specimens were observed under Olympus Microscope BH-2 at 40 × 10x magnification (numerical aperture 0.80) to count the mast cells recognized by red-purple metachromatic staining color on a blue background. Mast cells were counted in 20 fields, 4 sections per slide, and expressed as a total mast cell number, number of degranulating mast cells, and percentage of degranulated cells. Degranulating mast cells were defined as cells associated with ≥8 granules outside the cell membrane, as described previously. Analytes detected with ELISA include GM-CSF (#DY415-05; R&D), MCP-1 (#DY479-05; R&D), TNF-$\alpha$ (#DY410-05; R&D), and RANTES (#DY478-05; R&D) (Vincent et al, 2013; Cherukury et al, 2023).

## Statistical analysis

Data are shown as the mean ± SEM, and the sample size (N) is provided per condition in the associated figure legends. Trap length, pain-related behaviors, and vascular stasis were evaluated using regular two-way ANOVA with Tukey's post hoc multiple comparisons test; $P < 0.05$ was considered statistically significant, and groups being compared were identified in the associated figure legends. Pairwise comparisons for WBC counts, serum GM-CSF, cutaneous mast cells, and cytokines were evaluated using Student's unpaired two-tailed $t$ test; $P < 0.05$ was considered statistically significant. Data were analyzed using GraphPad Prism 10 software (GraphPad, Boston, MA).

## Acknowledgements

This work was supported by NIH Grants U18 EB029354 and R01s HL147562 and CA263806-01 and Susan Samueli Scholar Award to K Gupta; Diversity Supplement 3R01HL147562-03S to SB Kiven; and the University of California President's Postdoctoral Fellowship, A.P. Giannini Foundation Fellowship and Leadership Award, and K99 AT012494 to DA Argueta. We thank Dr Joni Ricks-Odie from the Institute for Clinical and Translational Science for statistical support. The content is solely the responsibility of the authors and does not necessarily represent the official views of the National Institutes of Health.

### Author Contributions

DA Argueta: data curation, formal analysis, validation, investigation, visualization, methodology, and writing—original draft, review, and editing.
H Tran: methodology.
Y Goel: formal analysis and methodology.
A Nguyen: methodology.
J Nguyen: methodology.
SB Kiven: investigation and methodology.
C Chen: investigation and methodology.
F Abdulla: funding acquisition, investigation, methodology, and writing—review and editing.

GM Vercellotti: conceptualization, funding acquisition, methodology, project administration, and writing—review and editing.
JD Belcher: conceptualization, formal analysis, funding acquisition, investigation, methodology, project administration, and writing—original draft, review, and editing.
K Gupta: conceptualization, formal analysis, funding acquisition, investigation, methodology, project administration, and writing—original draft, review, and editing.

## Conflict of Interest Statement

K Gupta reports grants from Novartis, Zilker, Grifols, 1910 Genetics, and Cyclerion and an honorarium from Novartis and CSL Behring outside the submitted work. DA Argueta reports honoraria from Cayenne Wellness Centers and Cyclerion outside the submitted work. GM Vercellotti and JD Belcher receive research funding from CSL Behring, Astellas/Mitobridge, Omeros, Hillhurst, and Sanofi, and are consultants with Astellas/Mitobridge and Octapharma. The remaining authors declare that the research was conducted in the absence of any commercial or financial relationships that could be construed as a potential conflict of interest.

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
