## [Reviewer comments · Life Science Alliance]

Life Science Alliance

Mast cell extracellular trap formation underlies vascular and neural injury and hyperalgesia in SCD

Donovan A Argueta, Huy Tran, Yugal Goel, Aithanh Nguyen, Julia Nguyen, Stacy B Kiven, Chunsheng Chen, Fuad Abdulla, Gregory M Vercellotti, John D Belcher and Kalpna Gupta

DOI: <https://doi.org/10.26508/lsa.202402788>

Corresponding author(s): Prof. Kalpna Gupta (University of California, Irvine)

Review Timeline:

Submission Date:	2024-04-22
Editorial Decision:	2024-06-04
Revision Received:	2024-08-16
Editorial Decision:	2024-08-22
Revision Received:	2024-08-26
Accepted:	2024-08-27

Transaction Report:

June 4, 2024

Re: Life Science Alliance manuscript #LSA-2024-02788-T

Prof. Kalpna Gupta
University of California, Irvine
101 The City Drive Swouth
Orange 92868

Dear Dr. Gupta,

Thank you for submitting your manuscript entitled "Mast cell extracellular trap formation underlies vascular and neural injury and hyperalgesia in sickle cell disease" to Life Science Alliance. The manuscript was assessed by expert reviewers, whose comments are appended to this letter. We invite you to submit a revised manuscript addressing the Reviewer comments.

Thank you for this interesting contribution to Life Science Alliance. We are looking forward to receiving your revised manuscript.

Sincerely,

B. MANUSCRIPT ORGANIZATION AND FORMATTING:

Reviewer #1 (Comments to the Authors (Required)):

In this manuscript, the authors demonstrated that the potential role of mast cells in the pathogenesis of sickle cell disease (SCD) related pain. They showed that mast cell extracellular trap (MCET) formation was dependent on the expression of peptidylarginine deiminase 4 (PAD4) and that inhibition of PAD4 led to a reduction in the formation of MCET and also SCD related pain cutaneous pain behavior. However, this benefit was not observed in SCD related deep/musculoskeletal pain (which is more is more in keeping with pain in SCD patients). Overall, the findings in the paper are in keeping with what has been reported prior from these investigators as well as by other investigators.

Some points that that authors should address include;

1. In the abstract, the words "monogenetic disorder" should be inserted after the words inherited.
2. The claim in the result section that tryptase granules wrapped around the nerve fiber, was not supported by any evidence in the result or from the literature. This statement should be revised to reflect the actual data presented.
3. Following from the above, the data presented shows that MCET interacted with the nerve fiber. There was no data presented to support the claim that MCET caused injury to the nerve fiber. Therefore, this claim in the manuscript is unsupported and should be revised or clearly presented as speculation on the part of the authors.
4. The authors claim that MCET formation mediated by PAD4 "underlies the persistence of chronic hyperalgesia in SCD" is also not supported by the data presented in the manuscript. This is even more so in light of the fact that PAD4 inhibition leading to the inhibition of MCET formation, did not attenuate deep/musculoskeletal pain, which is more pathognomonic of SCD pain, further undermine this conclusion. The data presented supports potential alleviation of cutaneous pain (less the issue in SCD).
5. It is not clear why the results in figure 2E were presented (not normalized to body weight) was presented differently from that of figure 3D (normalized to body weight). While the dosing of the various drugs was by body weight, it is not clear that benefit should be based on body weight, which suggested by the results in figure 3D.
6. The y-axis of one of the pair of figures in figure 3C is wrongly labelled.

Overall, this is an interesting paper, however it did not address a core pain phenotype that is more pathognomonic of sickle cell disease pain.

Reviewer #2 (Comments to the Authors (Required)):

The manuscript of Arguetta et al. identifies mast cell extracellular trap (MCET) release as an important actor on the process of hyperalgesia and inflammation in sickle mice. The use of GSK484, an inhibitor of PAD4, and therefore MCET release, improves hyperalgesia in vivo in HbSS mice and decreases inflammatory cytokines and mast cell degranulation in the skin.

The work performed by the authors is of great importance in the field of sickle cell disease research. Historically, the primary focus of therapeutic advancements has been on the prevention and treatment of vaso-occlusive crises and the underlying hematological abnormalities. Consequently, the chronic and debilitating pain experienced by patients has often been relegated to a secondary concern. This study brings much-needed attention to this critical aspect of sickle cell disease.

The experiments in this manuscript are very well designed to confirm the author's hypothesis. I believe that the manuscript has the potential to be considered for publication, taking in consideration the Life Science Alliance criteria for publication. However, some points still need to be addressed by the authors before publication:

- the authors should include experimental controls to demonstrate the efficacy of siRNA targeting FcER1 and TLR4 in mast cells (Fig 1E) and also illustrate the mast cell degranulation (Fig 4C) with the toluidine blue staining;
- the authors should also include experiments to demonstrate the effect of GSK484 in skin tryptase and MCET in vivo in HbSS mice treated with GSK484;
- would it be possible that the effects observed on WBC reflects decreased neutrophil counts? Did the authors access leukocyte populations in HbSS mice treated with GSK484?
- page 5 line 227: please correct Fig 5E to Fig 3E

Re: **Manuscript Number:** LSA-2024-02788-T

We deeply appreciate the reviewers' efforts in providing constructive suggestions. We have addressed the reviewers' concerns by performing several additional experiments, including, evaluating the effect of GSK484 on cutaneous tryptase release and MCET formation in sickle mice, etc. The revised text addressing the reviewers' comments has been highlighted in yellow in the manuscript and a point-by-point response to the reviewers' comments is provided below:

Reviewer #1

In this manuscript, the authors demonstrated that the potential role of mast cells in the pathogenesis of sickle cell disease (SCD) related pain. They showed that mast cell extracellular trap (MCET) formation was dependent on the expression of peptidylarginine deiminase 4 (PAD4) and that inhibition of PAD4 led to a reduction in the formation of MCET and also SCD related pain cutaneous pain behavior. However, this benefit was not observed in SCD related deep/musculoskeletal pain (which is more is more in keeping with pain in SCD patients). Overall, the findings in the paper are in keeping with what has been reported prior from these investigators as well as by other investigators. Overall, this is an interesting paper, however it did not address a core pain phenotype that is more pathognomonic of sickle cell disease pain.

Response to Reviewer #1: We deeply appreciate the constructive suggestions of the reviewer. We provide detailed responses to each of the concerns below and have revised the manuscript following the reviewer's suggestions.

Major comments:

Comment 1: In the abstract, the words "monogenetic disorder" should be inserted after the words inherited.

Response 1: The abstract text has been revised as follows:

Line 41: "Sickle cell disease (SCD) is the most common inherited monogenetic disorder."

Comment 2: The claim in the result section that tryptase granules wrapped around the nerve fiber, was not supported by any evidence in the result or from the literature. This statement should be revised to reflect the actual data presented.

Response 2: We show tryptase, a marker of mast cell activation, in red around blue nerve bundles in HbSS mice treated with vehicle in Figure 1A middle panel suggestive of mast cell activation in the vicinity of nerve fibers. We have also provided references showing mast cell activation in other models of pain and their proximity with nerve fibers. Our revision is as follows:

Lines 123-129: “We observed co-staining of tryptase granules (red) in close proximity with nerve bundles (cyan) suggestive of mast cell activation. This is similar to earlier findings showing dense clusters of tryptase granules intercepting the nerve bundles in the skin of HbSS-BERK mice (Gupta & Harvima, 2018). Our observations are complemented by other models of pain including mast cell degranulation evoked by glyceryl trinitrate induced headache in rats and mast cell accumulation at the site of incision near the nerve fibers in the hind paw of mice in a post-surgical model of pain (Green et al., 2019; Pedersen et al., 2015).”

Comment 3: Following from the above, the data presented shows that MCET interacted with the nerve fiber. There was no data presented to support the claim that MCET caused injury to the nerve fiber. Therefore, this claim in the manuscript is unsupported and should be revised or clearly presented as speculation on the part of the authors.

Response 3: The text has been modified to represent it as a speculation and provide supportive references for an association with nerve injury as follows:

Lines 155-159: “We speculate that this interception of nerve fibers with mast cell tryptase may lead to neural injury which may contribute to neuropathic pain. It is likely that mast cell activation contributes to the peripheral neural injury with sprouting nerve fibers in the skin, myelin sheath instability in the sciatic nerve, and neurovascular permeability that occur constitutively in HbSS mice (Kohli et al., 2010; Sadler et al., 2019; Vincent et al., 2013).”

Comment 4: The authors claim that MCET formation mediated by PAD4 “underlies the persistence of chronic hyperalgesia in SCD” is also not supported by the data presented in the manuscript. This is even more so in light of the fact that PAD4 inhibition leading to the inhibition of MCET formation, did not attenuate deep/musculoskeletal pain, which is more pathognomonic of SCD pain, further undermine this conclusion. The data presented supports potential alleviation of cutaneous pain (less the issue in SCD).

Response 4: We have shown that inhibition of PAD4 over a 4-week period continued to decrease thermal and mechanical hyperalgesia, which does suggest that extracellular traps contribute to the persistence of chronic pain. In the revised Figure 1A-B we show that GSK484 inhibited MCETs in the skin of sickle mice. Together, these data suggest that ETs and hence PAD4 may contribute to mechanical and cold hyperalgesia. This has been stated more clearly and **as a suggestion** in the revised text as follows:

Lines 200-208: “These findings **suggest** constitutive PAD4 activity, leading to MCET formation *in vivo*, **may** contribute to the persistence of chronic mechanical and thermal hyperalgesia in SCD. Persistence of mechanical and cold allodynia and hyperalgesia are critical features of pain in SCD, which may drive central sensitization (Brandow & Panepinto, 2016; Ezenwa et al., 2018; Molokie et al., 2020). A multicenter study of ~300 SCD patients over a period of 3 years, reported that “colder temperatures exacerbated sickle cell pain” (Smith et al., 2009). Cold hypersensitivity is a life-threatening condition in children and suggested to be a cause for acute pain in SCD (Ivy et al., 2023). Mechanical hyperalgesia is another characteristic feature of SCD with increased sensitivity to touch and even wind speed (Nolan et al., 2008).”

The deep/musculoskeletal pain may be due to avascular necrosis, which may require therapies that restore the damaged bones. The text has been updated as follows.

Lines 196-198: “No changes were observed with musculoskeletal hyperalgesia (Fig 2E), which is likely arising from avascular necrosis that requires therapies to restore damaged bone tissue (De Franceschi et al., 2020).”

Comment 5: It is not clear why the results in figure 2E were presented (not normalized to body weight) was presented differently from that of figure 3D (normalized to body weight). While the dosing of the various drugs was by body weight, it is not clear that benefit should be based on body weight, which suggested by the results in figure 3D.

Response 5: We appreciate this suggestion to correct our oversight. Figure 2E has been revised to show grip force normalized to body weight, similar to Figure 3D.

Comment 6: The y-axis of one of the pair of figures in figure 3C is wrongly labelled.

Response 6: Thanks for this correction. The y-axis for Figure 3C has been corrected to indicate PWL in response to a heat stimulus.

Reviewer #2

The manuscript of Arguetta et al. identifies mast cell extracellular trap (MCET) release as an important actor on the process of hyperalgesia and inflammation in sickle mice. The use of GSK484, an inhibitor of PAD4, and therefore MCET release, improves hyperalgesia in vivo in HbSS mice and decreases inflammatory cytokines and mast cell degranulation in the skin.

The work performed by the authors is of great importance in the field of sickle cell disease research. Historically, the primary focus of therapeutic advancements has been on the prevention and treatment of vaso-occlusive crises and the underlying hematological abnormalities. Consequently, the chronic and debilitating pain experienced by patients has often been relegated to a secondary concern. This study brings much-needed attention to this critical aspect of sickle cell disease.

The experiments in this manuscript are very well designed to confirm the author's hypothesis. I believe that the manuscript has the potential to be considered for publication, taking in consideration the Life Science Alliance criteria for publication.

Response to Reviewer #2: We thank the reviewer for their supportive comments. We have revised the manuscript with additional data as suggested.

Major Comments:

Comment 1: the authors should include experimental controls to demonstrate the efficacy of siRNA targeting FcER1 and TLR4 in mast cells (Fig 1E) and also illustrate the mast cell degranulation (Fig 4C) with the toluidine blue staining;

Response 1: The scrambled siRNA control condition for siRNA targeting of FcER1 and TLR4 in mast cells has been added to Figure 1F (previously Figure 1E) and representative images of mast cell degranulation visualized by toluidine blue staining have been added in Figure 4D.

Comment 2: The authors should also include experiments to demonstrate the effect of GSK484 in skin tryptase and MCET in vivo in HbSS mice treated with GSK484;

Response 2: We demonstrate the effect of GSK484 on cutaneous MCET formation and tryptase release and their interactions with peripheral nerve fibers and blood vessels in Figures 1A-B, and we have added additional text to the revised manuscript as follows:

Lines 132-135: “GSK484 treatment ameliorated MCET formation in the skin. Tryptase release and c-Kit expression are not observed/minimally expressed near the nerve fibers and blood vessels following GSK484 treatment of sickle mice (Fig 1A-B).”

Comment 3: would it be possible that the effects observed on WBC reflects decreased neutrophil counts? Did the authors access leukocyte populations in HbSS mice treated with GSK484?

Response 3: We evaluated the WBC counts but not neutrophils specifically. Ongoing investigation is examining the mast cell-neutrophil interactions, which will be presented in our future manuscripts.

Comment 4: page 5 line 227: please correct Fig 5E to Fig 3E

Response 4: We thank the reviewer for correcting our oversight. This has been corrected in the revised manuscript as follows:

Lines 237-239: “Pretreatment for 5 days with GSK484 or imatinib significantly reduced stasis over a 4-hour incitement period compared to vehicle-treated HbSS mice ($P < 0.05$ and $P < 0.01$, respectively, Fig 3E),...”

Sincerely,

Kalpna Gupta, Ph.D.
Professor of Medicine and Susan Samueli Scholar
Department of Medicine, Division of Hematology/Oncology,
University of California, Irvine
E-mail: kalpnag@hs.uci.edu

References

Brandow AM, Panepinto JA. (2016). Clinical Interpretation of Quantitative Sensory Testing as a Measure of Pain Sensitivity in Patients With Sickle Cell Disease. *J Pediatr Hematol Oncol*, 38(4), 288-293. doi:10.1097/MPH.0000000000000532

- De Franceschi L, Gabbiani D, Giusti A, Forni G, Stefanoni F, Pinto VM, Sartori G, Balocco M, Dal Zotto C, Valenti MT et al. (2020). Development of Algorithm for Clinical Management of Sickle Cell Bone Disease: Evidence for a Role of Vertebral Fractures in Patient Follow-up. *J Clin Med*, 9(5). doi:10.3390/jcm9051601
- Ezenwa MO, Molokie RE, Wang ZJ, Yao Y, Suarez ML, Dyal B, Abudawood K, Wilkie DJ. (2018). Differences in Sensory Pain, Expectation, and Satisfaction Reported by Outpatients with Cancer or Sickle Cell Disease. *Pain Manag Nurs*, 19(4), 322-332. doi:10.1016/j.pmn.2017.11.010
- Green DP, Limjunyawong N, Gour N, Pundir P, Dong X. (2019). A Mast-Cell-Specific Receptor Mediates Neurogenic Inflammation and Pain. *Neuron*, 101(3), 412-420.e413. doi:10.1016/j.neuron.2019.01.012
- Gupta K, Harvima IT. (2018). Mast cell-neural interactions contribute to pain and itch. *Immunol Rev*, 282(1), 168-187. doi:10.1111/imr.12622
- Ivy ZK, Belcher JD, Khasabova IA, Chen C, Juliette JP, Abdulla F, Ruan C, Allen K, Nguyen J, Rogness VM et al. (2023). Cold exposure induces vaso-occlusion and pain in sickle mice that depend on complement activation. *Blood*, 142(22):1918-1927. doi:10.1182/blood.2022019282
- Kohli DR, Li Y, Khasabov SG, Gupta P, Kehl LJ, Ericson ME, Nguyen J, Gupta V, Hebbel RP, Simone DA et al. (2010). Pain-related behaviors and neurochemical alterations in mice expressing sickle hemoglobin: modulation by cannabinoids. *Blood*, 116(3), 456-465. doi:10.1182/blood-2010-01-260372
- Molokie RE, Wang ZJ, Yao Y, Powell-Roach KL, Schlaeger JM, Suarez ML, Shuey DA, Angulo V, Carrasco J, Ezenwa MO et al. (2020). Sensitivities to Thermal and Mechanical Stimuli: Adults With Sickle Cell Disease Compared to Healthy, Pain-Free African American Controls. *J Pain*, 21(9-10), 957-967. doi:10.1016/j.jpain.2019.11.002
- Nolan VG, Zhang Y, Lash T, Sebastiani P, Steinberg MH. (2008). Association between wind speed and the occurrence of sickle cell acute painful episodes: results of a case-crossover study. *Br J Haematol*, 143(3):433-8. doi: 10.1111/j.1365-2141.2008.07354.x
- Pedersen SH, Ramachandran R, Amrutkar DV, Petersen S, Olesen J, Jansen-Olesen I. (2015). Mechanisms of glyceryl trinitrate provoked mast cell degranulation. *Cephalalgia*, 35(14), 1287-1297. doi:10.1177/0333102415574846
- Sadler KE, Lewis TR, Waltz TB, Besharse JC, Stucky CL (2019). Peripheral nerve pathology in sickle cell disease mice. *Pain Rep*, 4(4), e765. doi:10.1097/PR9.0000000000000765
- Smith WR, Bauserman RL, Ballas SK, McCarthy WF, Steinberg MH, Swerdlow PS, Waclawiw MA, Barton BA; Multicenter Study of Hydroxyurea in Sickle Cell Anemia. (2009). Climatic and geographic temporal patterns of pain in the Multicenter Study of Hydroxyurea. *Pain* 146(1-2):91-8. doi:10.1016/j.pain.2009.07.008
- Vincent L, Vang D, Nguyen J, Gupta M, Luk K, Ericson ME, Simone DA, Gupta K. (2013). Mast cell activation contributes to sickle cell pathobiology and pain in mice. *Blood*, 122(11), 1853-1862. doi:10.1182/blood-2013-04-498105

August 22, 2024

RE: Life Science Alliance Manuscript #LSA-2024-02788-TR

Prof. Kalpna Gupta
University of California, Irvine
101 The City Drive Swouth
Orange 92868

Dear Dr. Gupta,

Thank you for submitting your revised manuscript entitled "Mast cell extracellular trap formation underlies vascular and neural injury and hyperalgesia in SCD". We would be happy to publish your paper in Life Science Alliance pending final revisions necessary to meet our formatting guidelines.

- please be sure that the authorship listing and order is correct
- please add ORCID ID for corresponding author-you should have received instructions on how to do so
- please add the Twitter handle of your host institute/organization as well as your own or/and one of the authors in our system

A. FINAL FILES:

B. MANUSCRIPT ORGANIZATION AND FORMATTING:

****It is Life Science Alliance policy that if requested, original data images must be made available to the editors. Failure to provide**

original images upon request will result in unavoidable delays in publication. Please ensure that you have access to all original data images prior to final submission.**

The license to publish form must be signed before your manuscript can be sent to production. A link to the electronic license to publish form will be available to the corresponding author only. Please take a moment to check your funder requirements.

Sincerely,

August 27, 2024

RE: Life Science Alliance Manuscript #LSA-2024-02788-TRR

Prof. Kalpna Gupta
University of California, Irvine
101 The City Drive Swouth
Orange 92868

Dear Dr. Gupta,

Thank you for submitting your Research Article entitled "Mast cell extracellular trap formation underlies vascular and neural injury and hyperalgesia in SCD". It is a pleasure to let you know that your manuscript is now accepted for publication in Life Science Alliance. Congratulations on this interesting work.

DISTRIBUTION OF MATERIALS:

Again, congratulations on a very nice paper. I hope you found the review process to be constructive and are pleased with how the manuscript was handled editorially. We look forward to future exciting submissions from your lab.

Sincerely,
